# Dissecting Tumor Growth: The Role of Cancer Stem Cells in Drug Resistance and Recurrence

**DOI:** 10.3390/cancers14040976

**Published:** 2022-02-15

**Authors:** Beatrice Aramini, Valentina Masciale, Giulia Grisendi, Federica Bertolini, Michela Maur, Giorgia Guaitoli, Isca Chrystel, Uliano Morandi, Franco Stella, Massimo Dominici, Khawaja Husnain Haider

**Affiliations:** 1Division of Thoracic Surgery, Department of Experimental Diagnostic and Specialty Medicine–DIMES of the Alma Mater Studiorum, University of Bologna, G.B. Morgagni-L. Pierantoni Hospital, 47121 Forlì, Italy; franco.stella@unibo.it; 2Thoracic Surgery Unit, Department of Medical and Surgical Sciences, University of Modena and Reggio Emilia, 41124 Modena, Italy; valentina.masciale@unimore.it (V.M.); uliano.morandi@unimore.it (U.M.); 3Division of Oncology, Department of Medical and Surgical Sciences for Children & Adults, University of Modena and Reggio Emilia, 41124 Modena, Italy; giulia.grisendi@unimore.it (G.G.); bertolini.federica@aou.mo.it (F.B.); mauer.michela@aou.mo.it (M.M.); giorgia.guaitoli@unimore.it (G.G.); chrystel.isca@libero.it (I.C.); massimo.dominici@unimore.it (M.D.); 4Sulaiman AlRajhi Medical School, Al Bukayriyah 51941, Saudi Arabia; kh.haider@sr.edu.sa

**Keywords:** cancer, cancer stem cells, metastasis, microenvironment, stem cells, tumorigenesis, drug resistance

## Abstract

**Simple Summary:**

Cancer is one of the most debated problems all over the world. Cancer stem cells are considered responsible of tumor initiation, metastasis, drug resistance, and recurrence. This subpopulation of cells has been found into the tumor bulk and showed the capacity to self-renew, differentiate, up to generate a new tumor. In the last decades, several studies have been set on the molecular mechanisms behind their specific characteristics as the Wnt/β-catenin signaling, Notch signaling, Hedgehog signaling, transcription factors, etc. The most powerful part of CSCs is represented by the niches as “promoter” of their self-renewal and “protector” from the common oncological treatment as chemotherapy and radiotherapy. In our review article we highlighted the primary mechanisms involved in CSC tumorigenesis for the setting of further targets to control the metastatic process.

**Abstract:**

Emerging evidence suggests that a small subpopulation of cancer stem cells (CSCs) is responsible for initiation, progression, and metastasis cascade in tumors. CSCs share characteristics with normal stem cells, i.e., self-renewal and differentiation potential, suggesting that they can drive cancer progression. Consequently, targeting CSCs to prevent tumor growth or regrowth might offer a chance to lead the fight against cancer. CSCs create their niche, a specific area within tissue with a unique microenvironment that sustains their vital functions. Interactions between CSCs and their niches play a critical role in regulating CSCs’ self-renewal and tumorigenesis. Differences observed in the frequency of CSCs, due to the phenotypic plasticity of many cancer cells, remain a challenge in cancer therapeutics, since CSCs can modulate their transcriptional activities into a more stem-like state to protect themselves from destruction. This plasticity represents an essential step for future therapeutic approaches. Regarding self-renewal, CSCs are modulated by the same molecular pathways found in normal stem cells, such as Wnt/β-catenin signaling, Notch signaling, and Hedgehog signaling. Another key characteristic of CSCs is their resistance to standard chemotherapy and radiotherapy treatments, due to their capacity to rest in a quiescent state. This review will analyze the primary mechanisms involved in CSC tumorigenesis, with particular attention to the roles of CSCs in tumor progression in benign and malignant diseases; and will examine future perspectives on the identification of new markers to better control tumorigenesis, as well as dissecting the metastasis process.

## 1. Introduction

Cancer is considered one of the leading causes of death worldwide. Solid tumors are generally treated with surgery or medical approaches [1,2]. Treatment approaches include immunotherapies, chemotherapies, and radiotherapies [3,4,5,6]. One of the main problems with cancer is recurrence, which also greatly increases the mortality rate [3,7]. This is particularly dangerous because symptoms are frequently silent until the disease has significantly advanced. Even with curative resection, the percentage of recurrence remains high, nearly 30–70% [2]. This is attributed to symptoms that are frequently silent until the disease has advanced the residual cells enough to spread the tumor [8,9]. Recent research has identified and isolated cancer stem cells (CSCs), which are considered one of the primary causes of resistance to oncological treatments, and contribute to local and distant recurrence [10,11]. These CSCs are characterized by their ability to self-renew and their capacity to proliferate and contribute to significant tumor progression. The theory behind the formation of these cancer cells is exciting, as scientists believe that explaining this principle will enhance our understanding of cellular and molecular biology by deciphering their underlying mechanisms [12,13].

The CSC theory of tumor progression has been explained as a hierarchy, with cells at the top rank having the extreme capacity to self-renew and differentiate in a bulk population where the cells have the limited proliferative capacity [14,15,16,17]. CSCs share several properties with normal stem cells, such as their unlimited proliferative potential and self-renewal ability; however, their unique molecular markers have not been well-defined [18,19]. The scientific community must resolve this missing information in order to develop precision medicine in oncology [20,21,22]. Currently, efforts are underway to characterize the molecular profile of CSCs in order to better target them.

Strategies to target CSCs are considered important for analyzing the evolution of cancer therapy and the future of therapeutic approaches [23,24]. However, it is still challenging to identify and understand this cell subpopulation, which fosters tumor development and progression. The definition of their specific markers is the first hurdle to exposing them to targeted therapy. One of the most recently considered markers for CSCs is aldehyde dehydrogenase (ALDH), which shows higher metabolic activity in CSCs than the naïve cancer cells, and this peculiarity is exploited to identify them [25,26]. Sullivan et al. (2010) were the first to identify and report lung CSCs in primary human cell cultures through a cell-sorting technique specific to ALDH [27]. These data revealed elevated Notch pathway transcript expression in the ALDH high+ cells. In vitro and in vivo, multiple studies have been performed to show that ALDH high+ cells formed a significant population of self-renewing NSCLC stem-like cells with a high tumorigenic contribution. More recently, Masciale et al. [28] have identified and isolated lung CSCs from primary tumors based on ALDH expression, demonstrating their high capacity to form in vitro tumor spheres. Additionally, the results in adenocarcinoma and squamous cell carcinoma of the lung showed that the cells successfully grew for three weeks in a serum-free medium, forming spheres and overexpressing stemness genes, i.e., SOX2 and NANOG [28,29,30,31].

However, there is still a significant gap in targeting CSCs, as we still lack a specific or even a superficial marker that may be more suitable for future targeting of these cells [32,33]. Recently, CD44 and EpCAM antigens have been investigated in lung CSCs [31]. The data show high similarity with the ALDH high+ cells with which they were compared. The importance of identifying a unique and superficial marker lies in the capacity to selectively kill these cells, which have a vital role in cancer recurrence. When transplanted into immunodeficient mice, the data showed their capacity to regenerate the tumor [34,35].

Characteristically, CSCs can maintain their undifferentiated state through self-renewal, coupled with a high differentiation potential, thus allowing the maintenance of a stem cell pool and the generation of a heterogeneous progeny of differentiated tumor cells to constantly regenerate the tumor [36,37,38,39,40,41,42,43,44,45,46].

In 2019, Olmeda et al. defined three subpopulations of cells: cancer stem cells (CSCs), differentiated cells (DCs), and other cell populations [47,48,49]. Each cell type is created by a system based on molecular pathways and interactions [47]. Here, they demonstrated a sort of equilibrium by which tumor eradication or a possible relapse could possibly be derived. The scientific community has not yet been able to clarify these systems at this time, so there is a clear lack of well-defined criteria that support or hinder tumor development.

However, CSCs seem to be the crucial factor in tumor survival and dissemination, as they persist within the tumor microenvironment [50], which is described as a driver of the heterogeneity, plasticity, and evolution of the CSC subpopulation, also supporting niche formation, which is essential for CSC maintenance [47,48,49]. Notably, in solid tumors, the biomechanical properties of the microenvironment have recently been discussed for their capacity for inducing cancer stress, of stiffness around the network tissue, and abnormal interstitial fluid pressure (IFP) [23,24,25]. These combinations of factors may contribute to the metastatic process through the formation of “stressor foci,” which, driven by immunological distress associated with cellular collisions, induce the dissemination of tumor cells and CSCs from the vessels to the circulatory system [27,28]. These biological processes are responsible for tumor growth and recurrence, in association with the development of resistance to common cancer treatments.

## 2. CSCs in Tumor Growth and Dissemination: The Quiescent and the Active State

Quiescence is a unique state of a cell, wherein it can recover the ability to re-enter the cell cycle in response to different stimuli [51,52]. Adult stem cells are generally in a quiescent state, and they can be activated when needed, as observed during wound healing after tissue injury. Quiescent stem cells can respond to stimuli from their niche by activating molecular and transcriptional mechanisms to enter the cell cycle [53].

The quiescent state is responsible for recurrence in many solid cancers [54,55,56]. This peculiar capacity is also shared by CSCs, which are quiescent in the G0 state, lacking active cell replication and metabolic activity. This protects them against treatment, leading to the troublesome resistance of several types of cancer. It should be noted that eliminating the proliferative cells of the tumor via chemotherapy stimulates quiescent CSCs to activate and drive further tumorigenesis (Figure 1) [57].

To address this challenge, quiescent CSC populations should also be targeted by forcing them back into the cell cycle and exposing them to standard therapeutic intervention [58]. However, this is still complicated, since quiescent cells are challenging to identify, isolate, and study, even if it is necessary to expose them to standard therapeutic intervention. The reversal of a cell to a quiescent state may be prompted by alterations in its metabolism or by stress introduced into the tumor microenvironment. This phenomenon has also been described in adult stem cells, including hematopoietic, muscle, or neural stem cells [59]. CSCs can be quiescent for several years, but once activated, they can induce tumor progression and metastasis.

The mechanisms behind this aspect have not yet been completely defined, although several intrinsic mechanisms controlling gene expression seem to be involved in the switch to CSCs inducing cancer progression [60,61]. In particular, cyclin-dependent kinase inhibitors (CDKIs) play an essential role in the quiescence of adult stem cells. For example, CDKI1C is highly expressed in adipose-derived stem cells, and, through the downregulation of CDK2-cyclin E1, induces the cell to re-enter G0 [61,62]. Similarly, the tumor suppressor protein p53 is crucial for preserving genomic integrity. The action of p53 plays a critical role in maintaining genomic integrity by controlling cell cycle activity and inducing apoptosis [63]. It is also crucial for stem cells’ quiescence regulation, reducing or suppressing self-renewal and proliferation. Cell-cycle arrest by p53 is mainly mediated by the transcriptional activation of p21/WAF1, binding the cyclin E/Cdk2 and cyclin D/Cdk4 complexes to cause G1 cell cycle arrest [64,65].

Moreover, tumor suppressor retinoblastoma protein (RB), which represents a gatekeeper for the G1/S transition and blocks cell division, is dysregulated in several tumors [65]. The mechanism behind this protein is its capacity to accelerate cell cycle re-entry by binding and inhibiting transcription factors, i.e., E2F.; which is a downstream effector of the retinoblastoma (RB) protein pathway. Therefore, RB is believed to play a crucial role in cell division control [66]. Inactivation of RB protein induces E2F-mediated activation of cytokine signaling 3, which abolishes the cells’ quiescence [66].

The Notch signaling pathway is a molecular mechanism involved in stem cell differentiation and proliferation, which induces stem cells to enter into a quiescent state. The Notch mechanism can negatively regulate cell differentiation in skeletal muscle, by inducing the production of extracellular factors in the stem cells’ niche in order to preserve the quiescent state [67,68]. Regarding neural stem cells, Notch has been demonstrated to induce miR-708 expression in the quiescent state, and to repress the expression of focal-adhesion-related protein Tensin3, which can inhibit cell activation in the quiescent state.

However, other mechanisms such as DNA methylation, histone modifications, miRNAs, and long non-coding RNAs (lncRNAs) play a crucial role in controlling cell dormancy [69,70]. Recently, even metabolic alterations have been considered one of the main components in the quiescent cell state as it occurs in neural stem cells. Lipid anabolism of the neural stem cell seems to be related to their quiescent state.

It is hard to target CSCs or CSC-rich tumors, because these carcinoma stem cells can exit the cell cycle and not proliferate, therefore not inducing angiogenesis or active suppression of the immune system [71]. For some cancers, such as those of the breast, prostate, and kidney, this period of dormancy can last for many years, even decades, after apparently successful cycles of initial therapy. From a clinical point of view, patients with dormant metastatic cells are considered to have minimal asymptomatic residual disease [72]. Hence, this implies that understanding the mechanism of dormancy is of utmost clinical importance, as dormancy represents a critical time window during which treatments aimed at the elimination of the proliferative cells may succeed in preventing the relapse of the tumor. CSCs that have spread before surgical removal of the primary tumor may also persist in distant tissue environments as dormant cells within their niches. Patients with these quiescent reservoirs of CSCs have an increased risk for metastatic recurrence [73]. There are specific biochemical signaling pathways to maintain such a dormant state, including signals from the microenvironment, such as CXCL12, that activate AKT to promote survival, or reduced integrin-mediated mitogenic signaling, along with the actions of specific cytokines that induce quiescence associated with an ERKlow/p38high signaling state [71,74]. Thrombospondin-1 (TSP1), present in the basement membrane surrounding mature blood vessels, also promotes quiescence [75]. In addition, dormant cells can evade detection by NK cells through repression of NK activation ligands, and are likely subject to surveillance by the adaptive immune system, which can maintain tumor cells in a dormant state through the actions of IFNγ [76,77]. There is another peculiar characteristic of CSCs that can give them the ability to be quiescent: plasticity. Cell plasticity allows for cellular switching in response to signals from the surrounding environment, without the need for genetic change, even modulating cell cycle entry and exit [57,78]. This mechanism lets cancer cells adapt better to circumstances, especially adverse conditions, and allows CSCs to facilitate cancer progression, remaining safe and quiescent for years, ready for awakening as soon as the situation requires rebuilding the tumor itself [79,80].

Currently, cell plasticity represents a significant problem in cancer therapy, as it is involved in the evasion of treatment due to either incomplete response or resistance to repeated exposure to treatment [79]. Cell plasticity allows for cellular switching in response to signals from the surrounding environment without the need for genetic change. This mechanism lets cancer cells adapt better to circumstances and facilitate cancer progression [80]. This represents a significant challenge in CSC research, due to their nature as cells with uncontrolled proliferation activity and self-renewal ability, thus making it hard to target the tumor. Consequently, CSCs and all undifferentiated cells within a tumor may develop these properties, depending on the cell’s environmental context or under appropriate stimuli [35,81,82]. Undeniably, the tumor microenvironment plays a predominant role in this cell transformation, and provides a niche where the newly formed CSCs have all the necessary supplies for their growth and maintenance [43].

Further, stem-like cells within solid tumors enormously aid and accelerate the transformation process. Thus, the discovery of stem-like cells in human cancers has suggested a central role in tumorigenesis, due to their experimentally well-defined ability to seed new tumors [83,84]. For example, in breast cancer, the malignant transformation of stem-like cells had a significantly greater aggressiveness than the differentiated epithelial cells, since cell status strongly influences the behavior of progeny and subsequent oncogenic transformation [36]. Several cancer models have been proposed to demonstrate that factors secreted from the extracellular matrix are also responsible for the transition from CSC-like to CSC phenotype [85,86,87]. Both in vitro and in vivo studies have been conducted, showing that, among others, Interleukin 6 (IL-6) released by the breast and prostate CSCs are responsible for maintaining a dynamic equilibrium between the cells with non-cancer stem cells (non-CSCs) [88]. Thus, the generation of CSCs from non-CSCs is the outcome of cell plasticity, especially after a wound healing or a transformation in response to oncologic treatment [89]. Depending on their respective microenvironmental input, both CSCs and non-CSCs are plastic and tumorigenic, achieving the capacity for self-renewal [37]. This data is significant, and relevant for planning therapeutic strategies, as positive outcomes will be achieved only when both non-CSCs and CSCs are targeted. Indeed, it has been shown that under specific circumstances, non-CSCs can re-constitute the CSC population [38].

De-differentiation is a process by which non-CSCs have the chance to de novo form a tumor by replacing lost CSCs or tumor cells [37]. For example, breast cancer cells may acquire CSC-like properties after exposure to an adipokine secreted from mammary adipose tissues [39]. Moreover, there are also cases in which hypoxia-inducible factors (HIFs) damage the cells, driving the cellular switch from cancer cells to CSCs. One of the many consequences of low oxygen in the tumor environment, and HIF stabilization therein, is the induction of glycolytic enzymes and a shift from oxidative phosphorylation to glycolysis for energy production. This results in the enhanced production of metabolic acids, such as lactic acid, and acidification of the microenvironment, promoting CSC-like phenotypes [90,91]. These changes lead to the microenvironment signals, i.e., HIFs result in EMT that forces the cells to achieve the stem cell state, thus promoting the CSC phenotype and even metastasis. Increased expression of stem-cell-specific markers has been found in cancer cell lines from several solid tumors, such as prostate, brain, kidney, cervix, lung, etc., subjected to hypoxic conditions [92,93]. These findings are noteworthy, as they also hold implications for developing future anti-cancer therapies, as blocking these signals causes the cells to undergo the non-CSC to CSC transition [94].

It is also essential to consider the intrinsic phenotypic plasticity of stem-like cancer cells via spontaneous activation of one or more of the well-known pluripotency factors (OCT4, KLF4, c-MYC.; and SOX2) [95,96]. In this context, the role of the CSCs’ niche is crucial, since it regulates the transition between stem and non-stem cell states. CSCs are known for their capacity to escape death and metastasize after resting in a quiescent state for an extended period, protected by their niche [43]. Their niche contains a complex mixture of fibroblastic cells, immune cells, endothelial cells, perivascular cells and their progenitors, ECM components, and an intricate network of essential cytokines and growth factors [43]. In this complex microenvironment, one of the most critical cell types is the cancer-associated fibroblasts (CAFs), which play a primary role in maintaining the plasticity of CSCs through the promotion of tumor cell de-differentiation, the construction of a supportive niche for colonization formed by fibrils of collagens, and their ability to escape chemotherapy [43,97].

The CSC niche is a part of the tumor microenvironment (TME), representing the adjacent stroma together with the tumor cells. Non-CSCs in the tumor are also part of the CSCs’ niche. During the tumor’s development into a malignant condition, the state of CSCs within the tumor is of fundamental significance. It is, in turn, regulated by the TME and the CSCs’ niches within it. Cells in the CSCs’ niche produce factors that stimulate the CSCs maintenance/enrichment, angiogenesis, and the recruitment of the immune and stromal cells that secrete additional factors inducing tumor dissemination [44]. Recently, it has been shown that the signaling pathways of the cell cycle regarding growth factor secretion and stemness properties are specifically triggered to stimulate CSCs within the niche. In turn, cancer cells seem to participate in creating and preserving the niche [43]. The primary function of the niche is to protect CSCs from any possible damage incurred by hypoxia, cytotoxic T-lymphocytes, chemotherapy, and radiation therapy. During their spread, operated by the CSCs moving from the primary site to distant regions, only a small fraction of these disseminated cells may survive, inducing metastatic dissemination; in this context, the metastatic niche serves as a support for the tumorigenesis of metastatic stem cells, providing stromal cells, diffusible signals, and ECM components [34]. In particular, the concept of “stiffness” has a critical role in cancer, as one of the primary causes of inducing tumor growth. This mechanical stress made by cells and structures in the cellular environment seems to drive multiple behaviors, such as cellular morphogenesis, the epithelial-to-mesenchymal transition (EMT), and consequent cancer development and dissemination. Stiffness, in turn, is prompted by the accumulation of the extracellular matrix (ECM), since pressure can modulate cellular response to the activity of proliferation, differentiation, and migration [98,99]. In particular, signals controlling stiffness are regulated by the intracellular PI3K/Akt-mTOR-SOX2 pathway through the transmembrane protein integrin-β1 (ITGB1), with a high expression of CD133 and epithelial cell adhesion molecules (EpCAM) [100]. Experimental studies identified ITGB1 as strictly related to mechanical factors, cancer differentiation, and CSCs [101]. Schrader et al. have reported that hepatoma stem cells had higher clonogenic potential when cultivated on a soft matrix, since it allowed the cells to enter a quiescent state, thus improving their stemness [102].

In particular, 3D cell cultures allow modification of the stiffness of the substrate, modulating the different mechanical forces involved, which in turn affects the stemness of CSCs [102]. It is pertinent to mention that the continued increase in stiffness triggers the differentiation process of CSCs, thus maintaining the progression of the entire tumor. In addition, there are other stimuli, such as integrins, the presence of the epithelial receptor CD44, increased expression of SOX2, OCT4, and NANOG.; which induce non-CSCs to transform into CSCs through ECM modifications [98,99,100,101,102], and this inclines towards a poor prognosis. (Figure 2).

Besides these data, a large body of research describes the microenvironment fundamental for forming CSCs’ niche and their secretome that promotes abnormal neo-vasculature formation, and paracrine signals responsible for resistance to standard cancer treatments [103,104]. To develop effective new therapies, the possibility of targeting tissue stiffness and the ECM could represent an important goal in cancer treatment and should be evaluated.

## 3. Cancer Resistance: CSCs and the Tumor Microenvironment

The study of TME remains a point of prime interest and significance in solid tumors due to TME’s significant role, which may cause tumor resistance and progression. It has been divided into different components, visible by histopathological analysis, where the connections between normal and tumor tissue seem to predominate [105,106,107]. Forming niches through different cell types distributed in other tumor regions may generate a dynamic cancer environment.

Besides different tumor compositions, the overlap in signaling pathways and cell interactions can generate a substantial phenotypic repertoire within niches and CSCs that is more complex when the tumor is more aggressive. CSCs show different transcriptional and epigenetic signatures to maintain the niches [108,109,110,111,112,113,114,115,116].

CSCs can build an extensive solid network of connections between the tumor and the normal tissue, defining a specific role for each niche, easily linked by their dependence relationships. A particular niche usually develops under hypoxic conditions, which is the real driver for mediators of stemness [117]. CSCs can survive because of their high metabolism and affinity with nutrients, such as glucose, which promotes migration and dissemination of the tumor, inducing hypoxia and necrosis [117,118]. Moreover, CSCs induce the synthesis of angiogenetic factors and the formation of new vessels [119], and they are supported by the structures and signals coming from normal tissue, such as the CAF and niche ECM [120,121,122,123] (Figure 3).

Another interesting aspect that needs to be investigated is the connection and interaction between immune cells and CSCs in the potential development of new treatments that specifically target CSCs and immunity. Masciale et al. identified and isolated CSCs based on ALDH activity, and analyzed the tumor-infiltrating T-lymphocytes (TILs) of non-small cell lung cancer (NSLC) patients to observe the relationship between CSCs and TILs [28]. Data from 12 patients showed a positive correlation between CSCs and CD3+ cells and a stronger correlation between CSCs and cytotoxic CD8+ T-lymphocytes, thus suggesting a close interaction between the two cell populations [124]. These data indicate that CD8+ T cells could be crucial for a cell-mediated anti-tumor immune response via T-cell receptors binding with CSCs’ antigen. Unraveling these experimental data, a possible explanation for the positive correlation between CSCs and cytotoxic CD8+ T cells may be that the CSCs stimulate the immune system, triggering an immune response, particularly the CD8+ T cells, to suppress CSCs [124].

Understanding of the intricate interaction between T-lymphocytes and CSCs within the tumor would help develop novel combined treatment approaches. These could optimize the clinical benefit of current immunotherapies by interrupting the underlying mechanisms of tumor cell immune evasion [125,126,127,128]. In particular, scientists have recently given attention to the cytokines produced by immune cells, which seem to induce CSC maintenance and growth [129,130]. These novel discoveries have led to new approaches, such as single-cell genomics, epigenomic technology, and 3D culture systems, which provide new opportunities for profound understanding of this interaction [131,132,133,134]. CRISPR-Cas9 and RNA interference screening have also provided new insights into in vivo dependence and niche-cell interactions [131,132]. Single-cell sequencing efforts and multiregional tumor studies have defined the compositions of different intratumoral cells [134]. Furthermore, CSCs’ genomic information is used to analyze tissue biopsies [28,135]. The authors identified and isolated CSCs via fluorescent-activated cell sorting (FACS) to investigate cell behavior in vitro and gene expression from the surgical tumor tissue of 22 NSCLC patients [136].

Several other studies isolating CSCs in contrast defined a model in which individual tumors are composed of multiple subtypes of cells, implying that tumor microenvironmental diversity generates cellular heterogeneity [119,132]. Accordingly, therapies targeting a single niche have shown limited efficacy, since several components in the tumor microenvironment promote therapeutic evasion. The heterogeneity of tumors has been investigated intensively, and it seems to be related to intrinsic and extrinsic pathways [137,138]. It depends on the biological properties of cells, which can empower the tumor, though the extraneous features derive from the microenvironment and cell-to-cell interactions [139]. In this scenario, CSCs play a role as transformed cells, with the capacity to regenerate themselves, increase resistance to hypoxia for angiogenic stimulation, facilitate immune evasion, and increase cytokines and growth factor expression [44,140]. Recent studies have demonstrated that cancer cells can undergo de-differentiation and revert back to stem cell-like traits, such as self-renewal, growth, progression, and dissemination [141]. In this context, the TME is the pillar of preservation and diffusion of CSCs. The resistance mechanisms activated by CSCs evolve into tumor preservation and low response rate to common oncological treatments. Of course, our future capacity to target CSCs will improve with our understanding of the interaction between CSCs and TME [142,143,144,145]. The complexity of TME makes it challenging to understand its connection with CSCs, since several components contribute synergistically to stimulate and preserve CSCs’ growth and subsequent tumor dissemination [146].

For an in-depth understanding of CSCs and TME.; and their interaction, an in vivo study was performed through xenograft models of highly immune-compromised NOD-SCID/IL2g-/- (NSG) mice, in which the growth capacity of CSCs was compromised-immune-cell-dependent, especially B-lymphocytes and natural killers (NKs) [147,148]. The published data show that the absence of immune cells may directly or indirectly influence tumor growth and the presence of cancer-initiating cells (CIC) [149,150]. For this reason, an experimental immunocompromised mice model is considered a robust method to understand the dependence of CSCs on TME. One of the most practical approaches to studying this relationship is the engraftment of a primary cell culture enriched in CSCs into immunocompromised mice [148]. The use of an adequate number of CSCs remains an important aspect, as is a matrix that supports precise implantation of cells in the exact inoculation site. Matrigel is the most used enriched matrix in experiments [151,152].

The focus should be on fibroblasts within TME representing the “stromal bed” in the central part of the tumor. The stroma represents the essential component of TME.; as it plays a crucial role in reverting differentiated cells to de-differentiated phenotypes, from which the generation of CSCs takes place. Specifically, it drives cells’ plasticity through critical signal transmission, such as the Wnt and Notch pathways [153,154]. Another essential aspect being considered is the role of CAFs within TME.; for the secretion of growth factors and cytokines, such as platelet-derived growth factor (PDGF), and vascular endothelial growth factor (VEGF), which induce tumor progressions [154,155,156,157,158].

The regulation of an acidic and hypoxic microenvironment is often characterized by two primary oxygen controllers, HIF1A and HIF2A. These factors are susceptible to cellular pH modifications, as both hypoxia and pH changes may cause the metabolic switch into a more aggressive cancer cell phenotype through the glycolytic process and the induction of EMT. Additionally, other events promote tumor growth sustained by CSCs, such as the overexpression of C-X-C-chemokine receptors and the upregulation of gene expression of Snail and Twist [58,159]. The two transcription factors Twist and Snail are members of a family of EMT regulators, which induce metastasis by down-regulating E-cadherin. Their expression is also related to the β-catenin signaling pathway (Figure 3).

### TME Supports CSCs

Recent studies correlate expression of Snail and Twist with the loss of cell adhesion, increased cell migration, and accumulation of β-catenin signaling, which results in increased aggressiveness documented in, e.g., metastatic ovarian and breast carcinomas [160,161].

Furthermore, scientists have identified certain genetic and epigenetic factors playing a critical role in the concept of plasticity. It has recently been suggested that the metabolic reprogramming of cancer cells may represent a new aspect of cancer that redirects cancer cell status from non-CSC to CSC [162,163,164]. Intracellular metabolism sets cellular proliferation and differentiation [165], and new insights report that CSCs and their differentiated progeny may show different metabolic states [162,166]. CSCs undergo oxidative phosphorylation in breast cancer, although non-CSC cells preferentially carry out aerobic glycolysis [167]. However, tumors represent a mixture of cancer and microenvironmental cells communicating through a bidirectional metabolic flux, where every part influences each other in mutual metabolic reprogramming [168]. In this context, CAFs have a metabolic role in reprogramming cancer cells by inducing a reverse Warburg phenotype [167,168,169,170,171].

Tumor dissemination starts without clinical symptoms, allowing the disseminated cells to acquire a dormant state, which seems to reflect the resistance to therapies in advanced-stage tumors [172]. Since dormant cells may cause tumor recurrence, quiescence and slow growth are features of tissue-residing stem cells, and a pertinent question is whether CSCs may be the cause of metastatic dissemination [173]. MICs have been only recently demonstrated in solid tumors, such as breast cancer, colon cancer, and lung cancer [174,175,176,177]. It is interesting that MICs are found in CSC subpopulations [178]. Tumor dissemination needs an environment for the tumor to spread. The so-called ‘‘metastatic niche’’ may represent a native stem cell niche of the distant organ with stem cell properties [177,179,180].

To summarize, the CSCs’ niche is an active environment regulated by developmental signaling pathways, i.e., Wnt, Notch, and the chemokine CXCL12 [43], endothelial-mediated paracrine stimulation, ECM components, and the secreted enzymes, i.e., lysyl oxidase (LOX) [181]. Moreover, the release of inflammatory components, such as cytokines and enzymes, induces the primary source of the tumor in a “pre-metastatic niche” located in distant organs [182,183,184]. However, this state of quiescence derived from the tumor dormancy is due to reduced vascularization (representing angiogenic dormancy) and high cytotoxic activity in the immune system (immune-mediated dormancy) [185,186]. Finally, tumor cells may drive progression or tumor growth latency, depending on the presence of specific factors and cytokines in the surrounding microenvironment [182]. In particular, the mutations harbored by these cells maintain the integrity of the tumor [187], and it is now accepted that TME has an important role in forcing the genetic evolution toward some mutations favorable for cancer cell survival. Among other factors, TME is a promoter of the “clonal” choice that selects those cells to induce tumor development and maintenance. It is now well-established that CSCs and TME dynamically interact to influence each other, involving different cellular players [188,189,190,191].

## 4. The Molecular Mechanisms Switching on CSCs and Metastasis

A century ago, the theory of cell fusion between macrophages and tumor cells was considered the leading underlying cause of metastasis. Since then, other studies have provided evidence that cell fusion could lead to tumors and their metastasis [192]. The cell fusion process may result in two forms of hybrids, heterokaryons or synkaryons. In heterokaryons, the genetic information of the parental cells remains located in segregated nuclei, thus leading to the development of bi- or multi-nucleated hybrids. This was first observed in vitro in the Sendai virus in murine Erlich ascites cells combined with human HeLa as the fusion cells [193]. Changes in the morphological characteristics of both cells were observed. Specifically, somatic cells underwent rapid nuclear reprogramming and epigenetic modifications through fusion to form hybrid cells with distinct genetic and phenotypic characteristics compared to the parent cells [194]. In synkaryons, only one nucleus was formed due to the union of the two cell types [195]. Along with the discovery of CSCs, it has been shown that stem cells can fuse with differentiated cells, forming a heterokaryon, having the functions and characteristics of each of the two cell types involved in the fusion process. Cell fusion was described as a mechanism for generating CSCs by Gauck et al. [196]. The authors reported that the fusion between human breast epithelial cells and human breast cancer cells formed hybrid cells with specific CSC properties, such as the capacity for colony-forming spheres. In addition, another example describes the spontaneous formation of heterotypic hybrids between MSCs and lung cancer cells. The newly formed hybrid cells expressed the stem cell marker prominin-1 [197] alongside the expression of other stem cell-like phenotype characteristics, such as the transcription factors octamer-binding transcription factor 4 (OCT4), ALDH-1, B-lymphoma Mo-MLV insertion region 1 (BMI-1), and sex-determining region Y-box 2A (SOX-2A) [198]. Notably, these changes occur over a limited period compared to the genetic changes due to random mutations. Further, some biological processes, such as inflammation and hypoxia, could enhance cell fusion [199,200].

### 4.1. Horizontal Gene Transfers between Cells

Horizontal gene transfer (HGT) or lateral gene transfer (LGT) involves a process in which an organism transfers genetic material to another non-descendant cell [201]. Genomes of evolving cells are subjected to higher plasticity than most conserved cells, and tumor cells must continually reinvent themselves to propagate in the recipient organism [202]. It was hypothesized that circulating tumor DNA was propagated into the human body through biological fluids, and is inserted into normal stem cells, which could then be transformed into CSCs. The incorporated genes were expropriated for vertical inheritance [203,204].

### 4.2. Genetic Instability

As stem cells age, like any other body cell they can accumulate genetic mutations; however, the critical difference is that throughout their lifespan, incremental acquisition of mutations in the stem cell population and their progeny can give rise to cancer [205,206]. Knowing this, in-depth understanding of stem cell biology is a prerequisite for cancer researchers, because tumorigenesis proceeds via the accumulation of inherited acquired somatic mutations and epigenetic changes, which may modulate gene expression. Because of their specific ability to self-renew, which requires a high rate of cell division, stem cells are the most appropriate cell type to accumulate chromosomal abnormalities and stochastic mutations [207,208,209]. As stem cells divide, acquired mutations accumulate in the stem cell pool over time [210,211]. Any loss of functional genes due to mutation during the asymmetric cell division process regulates the fate of stem cell-derived daughter cells, and may lead to an uncontrolled self-renewal that disrupts stem cell homeostasis and ultimately leads to cancer [212].

### 4.3. Molecular Pathways in Cancer Stem Cells

CSCs are endowed with self-renewal and high proliferative potential, characterized by both symmetric and asymmetric cell divisions, akin to normal stem cells [213,214]. For this reason, the principal molecular mechanisms regulating CSCs are the same as that of normal stem cells, which regulate and coordinate embryonic development and tissue repair, particularly the Wnt, Hedgehog, and Notch pathways (Figure 4) [215].

#### 4.3.1. Wnt Signaling

The Wnt pathway was identified in the late 1990s as a proto-oncogene responsible for the development of tumors in transduced mice. Several pathways regulate Wnt signaling, with three arising as the most important, the “canonical” Wnt pathway (also called Wnt/β-catenin). The Wnt/β-catenin pathway is activated when a Wnt ligand binds to a seven-pass transmembrane Frizzled (Fz) receptor and its co-receptor, low-density lipoprotein receptor-related protein 6 (LRP6) or its close relative LRP5 [216,217]. The formation of the Wnt-Fz-LRP6 complex and the recruitment of the scaffolding protein Disheveled (Dvl) results in LRP6 phosphorylation and activation that leads to recruitment of the Axin complex to the receptors [218,219]. These events lead to the inhibition of Axin-mediated β-catenin phosphorylation, and thereby to the stabilization and accumulation of β-catenin in the nucleus and formation of complexes with TCF/LEF. These molecular events activate Wnt target gene expression, thus establishing the mitotic spindle, regulating asymmetric cell division, underpinning stem cell maintenance, and producing differentiated cells [220].

Recently, many studies have implicated Wnt signaling in CSCs of solid tumors, i.e., glioma, and adenocarcinoma of the colon, as an essential regulator of the tumor-initiating cells [221,222,223].

#### 4.3.2. The Hedgehog Signaling Pathway

The molecular mechanism of the Hedgehog (HH) pathway is initiated by the HH ligands binding to the Patched receptors, blocking the inhibition of Smoothened, a seven-transmembrane domain receptor, which is responsible for the activation of intracellular signal transduction via the glioma-associated oncogene (GLI) transcription factor [224]. This protein enters the nucleus and activates the target genes of the HH. This pathway plays a crucial role during organogenesis by mediating cell–cell communication. It also underlies the regulation of cell proliferation and EMT.; essential processes involved in carcinogenesis and subsequent tumor progression [225]. In addition, active HH signaling may also be a significant cause of cancer treatment failure in cancer patients, due to impaired chemotherapeutic drug responses or by actively inducing more aggressive and treatment-resistant tumors [226]. HH signaling is also associated with CSC identification in several solid tumors, such as breast cancer, glioma, basal cell carcinoma, gastric cancer, and colon carcinoma through the regulation of stemness-related genes, i.e., OCT4, SOX2, and BMI1 [99,100]. Moreover, HH is involved in regulating tumor spheroid formations, as observed in the case of glioblastoma (GBM) neurospheres, by controlling NANOG.; nestin, BMI1, and gene expression [227].

An experimental mouse model of NOD/SCID showed that the engraftment of tumor neurospheres pre-treated with cyclopamine (a drug that binds to the heptahelical bundle of Smoothened) blocked the HH signaling, thus resulting in tumor growth reduction [228]. These data demonstrated that inhibition of the HH pathway can prevent clonogenic growth and self-renewal of the GBM-derived CSCs (GSCs) [229]. Moreover, a combined treatment of cyclopamine and 10 Gy of radiation therapy showed a significant reduction in neurosphere growth. These data highlighted that HH blockade might affect CSCs, which generally are not targeted by chemotherapy and radiotherapy alone [230,231].

#### 4.3.3. The Notch Signaling Pathway

The Notch gene was first discovered in a *Drosophila melanogaster*, and its mammalian homolog has four receptors (Notch1–4) and five Notch ligands (Delta-like 1, 3, and 4, Jagged 1, and Jagged 2), which are transmembrane proteins regulating the communication between cells [232]. As a ligand binds with a Notch receptor, it unleashes a proteolytic cleavage of the Notch intracellular domain (NICD). This promotes translocation into the nucleus to bind with the specific transcription factor CSL [233]. The NICD/CSL transcriptional activation complex is responsible for the activation of the basic helix-loop-helix (bHLH) family of transcription factors such as HES.; HEY.; and HERP (HES-related repressor protein) [234]. HES and HERP are viewed as primary targets/effectors of Notch, as Notch signaling relies on close cooperation between HES and HERP.; which have distinctive repression mechanisms that regulate the mRNA of the target gene [235]. The dysregulation of Notch is related to many malignant tumors, as Notch acts as an oncogene and a suppressive gene, primarily depending on the environmental context and the cues involved there. For example, an upregulation of the Notch pathway is responsible for GBM and malignant medulloblastoma [236,237]. Hence, different methods for silencing Notch have been explored, such as inhibitor compounds, monoclonal antibodies, and siRNA. Co-inhibition of Notch and HH in an in vitro model of the GBM neurosphere showed a reduction in tumor growth and clonogenicity [238,239]. The CD133+ CSCs isolated from the glioma cell line were susceptible to γ-secretase inhibitors (GSI), or Notch1/2 knockdown, compared to the respective CD133-negative glioma cells [240,241]. This evidence highlights that Notch may be considered a promising target for developing more effective glioma therapies [242,243].

Moreover, as in ovarian cancer, CSCs are facilitated in migration and cell invasion through Notch1 even in the absence of hypoxia, which is usually a major factor supporting metastasis [244,245]. Indeed, Notch signaling is related to CSCs of various origins in solid and hematologic tumors, i.e., breast cancer, pancreatic cancer, colon carcinoma, and acute myeloid leukemia [246]. It has also been shown that the activation of Notch promotes cell survival and self-renewal, and inhibits apoptosis [247]. As described in breast cancer research, abnormal Notch signaling triggers CSCs to promote self-renewal and metastasis [248]. In particular, microRNA-34a is a suppressor of Notch1 gene expression, leading to an inhibition of cell proliferation activity and increases in the apoptotic processes of breast cancer cells and stem cells [249]. An important master gene regulator in breast cancer, such as BRCA1, activates the Notch pathway in breast cancer cells through transcriptional upregulation of Notch receptors and ligands [250]. Moreover, BRCA1 regulates JAG1 through a Delta Np63-dependent mechanism, whose role in stem cell fate is well known [251].

## 5. CSCs as Novel Targets for Cancer Therapy: New Perspectives to Control Tumorigenesis

Cancer therapy approaches are one of the most exciting areas of research. Despite the introduction of immunotherapy, which is responsible for significantly improved prognosis, the chances of recurrence and death remain very high. Gold standard treatment as per oncological guidelines comprises surgery for the early stages and chemotherapy/radiotherapy for locally advanced and generally advanced disease. CSCs seem to play a pivotal role in cancer recurrence [252,253]. The scientific community has deeply analyzed this aspect [254,255]. It has also highlighted the importance of targeting this sub-population of cells, since common oncological treatments are not entirely effective against CSCs, which can survive in a quiescent state and replicate after an injury, such as those triggered by chemotherapy. Hence, the need to target CSCs has led researchers to focus their attention on ALDH.; now being considered the best marker to identify and further target CSCs in several solid tumors.

To better understand the role of a previously validated cell cycle gene signature associated with cancer recurrence [136], Masciale et al. isolated CSCs from fresh surgical lung cancer specimens by isolating ALDH high + cells [28,136]. It should be noted that ALDH is not only a marker but a functional regulator of CSCs. ALDH is an enzyme of the ALDH superfamily known to regulate cellular functions related to self-renewal, expansion, differentiation, and resistance to drugs and radiation. Future treatment approaches may seek to discover a marker able to target CSCs selectively. In particular, a superficial marker is required to make it easier to develop new clinical treatments in cancer therapy. In this regard, an important achievement in lung cancer has recently been reached through the study of CD44+/EPCAM+ cell populations [31], which showed a high correlation and affinity with CSCs, previously identified by ALDH high cells. This could represent a breakthrough for lung cancer treatment, since it can bind and target CSCs through their surface proteins/markers.

Beyond the urge to find a surface marker, and beside the fact that several publications have shown the presence of stemness genes in CSCs, new strategies are being carried out, specifically targeting the activity of cancer stem cells, in particular on a CSC gene signature [136]. A cross-sectional study involving 22 patients undergoing surgery for adenocarcinoma or squamous cell carcinoma of the lung investigated a huge and already known panel of 31 cell-cycle genes related to cancer recurrence, for both early and locally advanced stages, to create more tailored therapies in the future [136]. The novelty of this recently published cross-sectional study was in identifying the same recurrence of genes in CSCs for early and locally advanced stages. In particular, further analysis has revealed that a subset of these genes is differentially expressed among stages, grouped as early in stage I-II and locally advanced with stage IIIA.; suggesting genes that were essential during the initial phase of the tumor and others which lead to metastasis. Moreover, stemness genes, such as OCT4, NANOG.; and SOX2 have higher expression in CSCs compared to non-CSCs [28]. A gene signature study investigated a possible RNA interference-mediated down-regulation of this gene expression, including anti-apoptotic genes, to ensure a more effective eradication of CSCs in the future. For example, in glioblastoma (GBM), inhibition of checkpoint kinase 1 (Chk1) and checkpoint kinase 2 (Chk2) activity decreased its resistance to radiotherapy [256]. L1 cell adhesion molecule (L1CAM) shRNA induced the elimination of CD133+ glioma cells, but it did not affect negative cells.

Along with this advanced approach, more typical though no less important approaches have been developed to reduce drug toxicity and chemo-resistance (Figure 5). In particular, targets related to ABC transporters have been studied as one of the main ways to reduce resistance to medical oncological treatments. Down-regulation of ABC transporters may inhibit drug efflux, causing the drug to persist longer within the tumor cells, including CSCs. This would benefit the tumor cells’ removal. The scientific community has shown that down-regulation of ABC transports should be standardized to prevent side effects [257]. Strategies to silence CSC-related genes that can reduce or inhibit their leading molecular roles, such as growth and self-renewal, have been studied in cervical cancer stem-like cells [258]. In tumor cells, blocking CSC signaling pathways, such as AKT and signal transducer and activator of transcription-3 (STAT-3), in glioma is an effective practical approach that needs further investigation for application in other solid tumors [259,260,261]. There is also the possibility of unique CSC niches, since they represent a continuous supply for these cells, preserving cancer.

A recent study has shown that CSCs create their niche around blood vessels, reducing radiotherapy efficacy. Notably, this has been described within cells from angiosarcomas (a rare vascular tumor) of the lung positive for ALDH.; which suggests a central role for ALDH in the angiogenetic process, since it has also been detected in the endothelial stem-like cells of these vascularized tumors [262]. Research provides examples of drugs that work against angiogenic factors affecting CSCs, such as one using an in vivo mouse model to study the effect of the vascular endothelial growth factor receptor 2 (VEGFR2) antibody that, in association with a chemotherapeutic treatment, was able to reduce CSCs’ subpopulation [181]. This aspect suggests that a therapeutic approach based on anti-angiogenesis can eradicate CSCs and represents a promising approach for developing new cancer treatments [181,263] (Figure 6).

Nanoparticles (NPs) have been developed to increase the targeted efficiency of drugs by increasing their stability, thereby facilitating their entry into the nucleus for a longer-lasting effect, allowing a reduction in dose and a possible decrease in adverse effects [264,265]. Still, their immunogenicity and uneven intratumoral distribution often restrict their therapeutic potential and clinical application. There is a dire need to use cellular vehicles with drug-loaded NPs. Therefore, combining nanomaterials with new nanotechnology-based drug delivery platforms, such as exosome-based approaches, could represent promising new tools [266,267]. Exosomes can be used as drug and miRNA delivery vectors in cancer treatment, as has been used in other cases [268]. Through their membrane-anchored ability, exosomes can be taken up by the cells via endocytosis to transfer their content, such as miRNAs and therapeutic proteins [269,270].

Compared with liposomal and metal or polymeric nanomaterials, exosomes can overcome the constraints of poor bioavailability and reduce off-target cytotoxicity and immunogenicity. In 2016, Kim [271] found that paclitaxel-loaded exosomes derived from macrophages, compared with paclitaxel-loaded liposomes, significantly increased cell uptake in vitro experiments using lung cancer cell lines. Li et al. [272] have modified the surface of the exosomes with a peptide-targeting mesenchymal-epithelial transition (MET) factor gene (c-Met), overexpressed on triple-negative breast cancer cell surfaces, with the result of improving the cellular uptake efficiency and antitumor efficacy of doxorubicin. Exosomes are also valuable in modulating CSCs by targeting CSC-specific signaling pathways, such as the Wnt, Notch, Hippo, Hedgehog, NF-κB.; and TGF-β pathways, which are extremely important for the self-renewal, differentiation, and tumorigenesis of CSCs.

Selective targeting of CSCs through these pathways using exosome-loading inhibitors (miRNAs or siRNAs) is considered a promising treatment approach [273]. For example, in lymphoma, the cells use Wnt signaling pathways to send information to neighboring cells via exosomes. Recent studies have demonstrated that exosomal Wnt from fibroblasts could induce tumor cell de-differentiation, triggering chemotherapy resistance in colorectal cancer cells and thus suggesting that interference with exosomal Wnt signaling could improve chemo-sensitivity for more effective treatments. Other studies using in vivo animal models showed the substantial and more potent effect of exosome-based chemotherapy than free drugs. Moreover, compared to free drugs, the exosome-based delivery platform may significantly reduce side-effects, while remaining much more effective in killing drug-resistant cancer cells.

A study of engineered exosomes containing miRNA (particularly miR-21) show they effectively downregulated target genes PDCD4 and RECK of the miR-21 in glioma cell lines [274]. Based on this data, it is possible to develop new strategies relying on engineered exosomes carrying tumor-suppressor proteins, nucleic acid components such as miRNAs, or targeted drugs functioning as precision medicine. Though progress has been made in this field, the molecular mechanisms of exosome production and their biological roles in tumor progression need further clarification [275]. Many more issues need to be addressed before this novel approach can become a clinical reality. However, exosome-based strategies in cancer treatment have shown great promise in experimental studies.

## Figures and Tables

**Figure 1 cancers-14-00976-f001:**
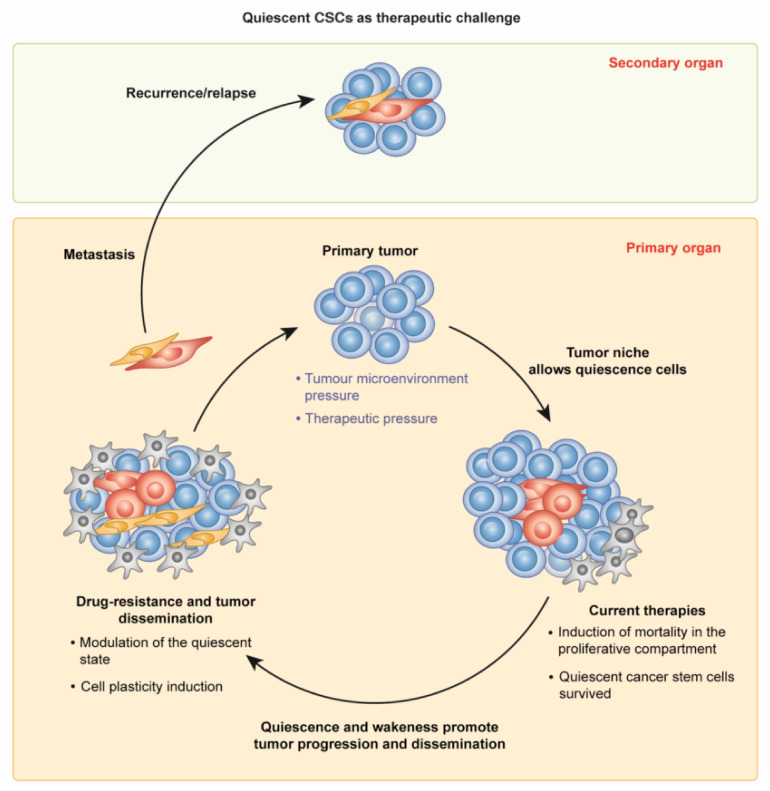
Quiescence in cancer stem cells. CSCs have the unique capacity to undergo a dormant state, making them invincible to external attack and preserving a reservoir of highly proliferative cells, which can recreate the entire tumor, if necessary.

**Figure 2 cancers-14-00976-f002:**
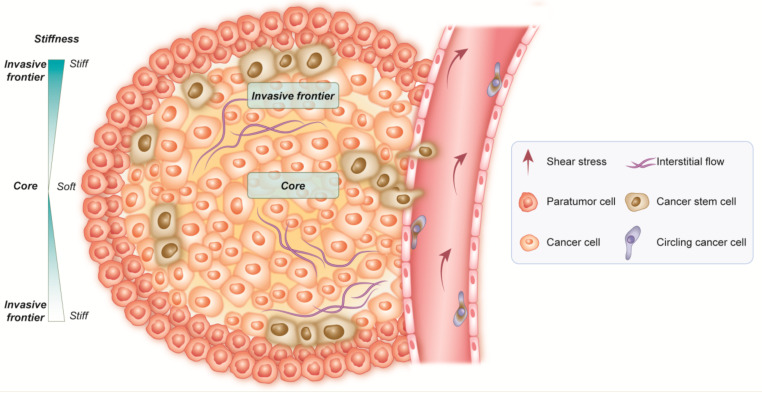
The role of stiffness in cancer. Stiffness triggers the differentiation process of CSCs, allowing the tumor to constantly reconstitute itself. A stiff matrix fosters CSC dissemination in the bloodstream, responsible for metastatic dissemination.

**Figure 3 cancers-14-00976-f003:**
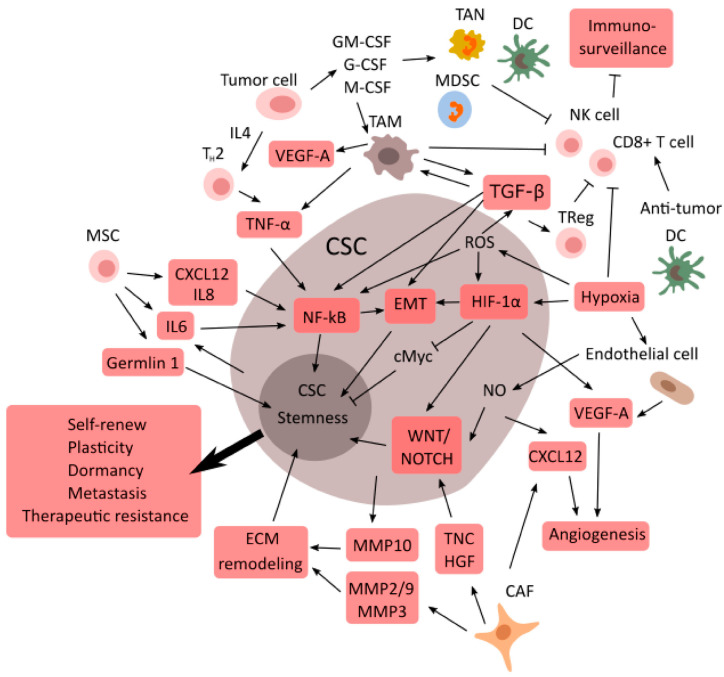
The tumor microenvironment supports CSCs. The tumor microenvironment is primarily responsible for the regulation of CSC plasticity, activating stemness pathways and promoting immune escape through cytokine release and inactivation of the T-lymphocytes, thereby inducing a tumor cell to acquire the CSC phenotype or a mesenchymal stroma cell to complete the epithelial mesenchymal transition towards cancer phenotype. TME prompts the angiogenetic de novo formation via CSC spread in the bloodstream for metastatic dissemination.

**Figure 4 cancers-14-00976-f004:**
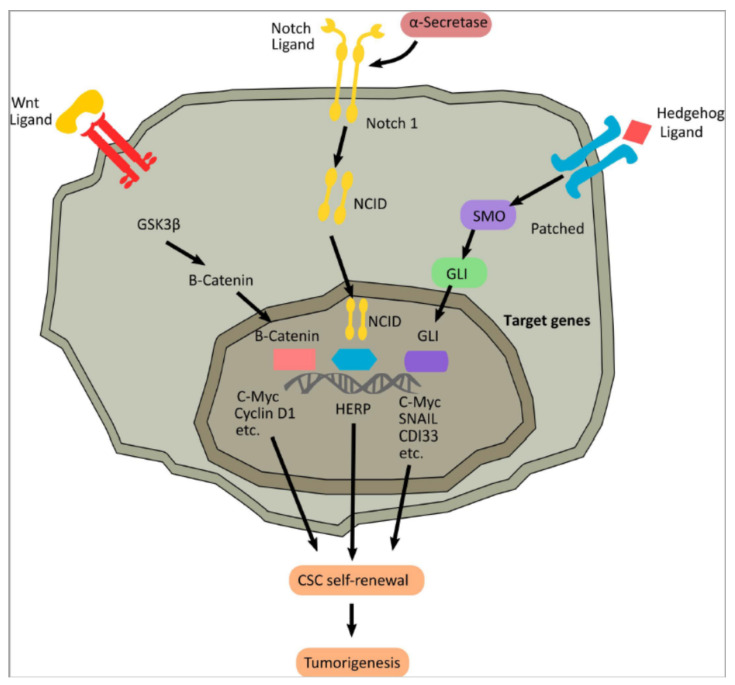
Signaling pathway regulating self-renewal in CSCs. Notch signaling, like the Wnt and Hedgehog pathways, is a highly evolutionarily conserved pathway of cell fate determination, with major relevance across multiple aspects of cancer biology, from angiogenesis and tumor immunity to the regulation of CSCs’ self-renewal ability.

**Figure 5 cancers-14-00976-f005:**
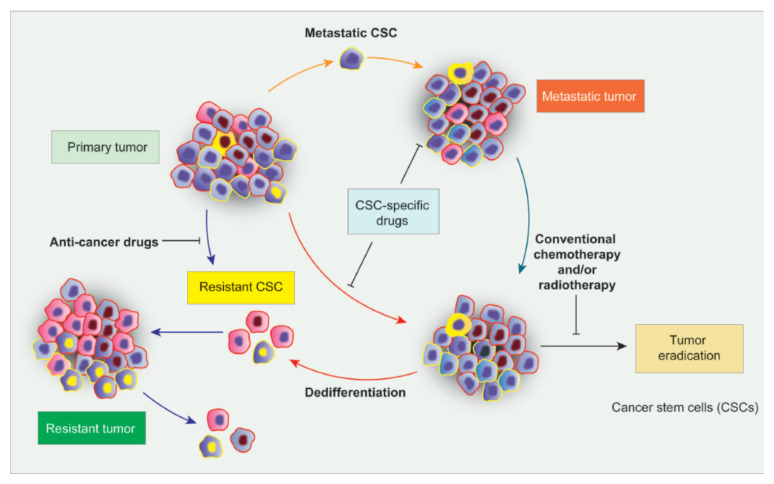
Challenging cancer drug resistance. A novel approach to cancer therapy focused on CSCs to break through the mechanism of drug resistance in cancer.

**Figure 6 cancers-14-00976-f006:**
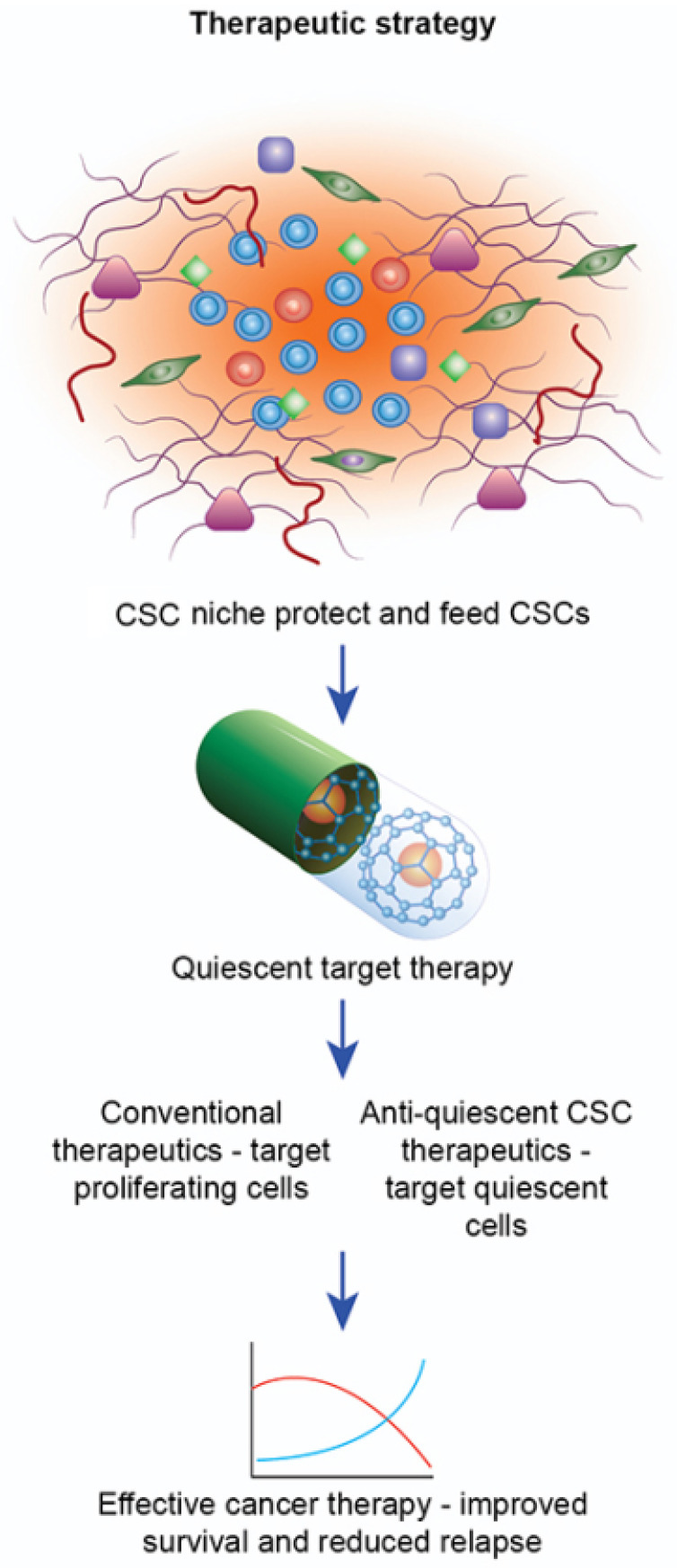
Target therapy specific to quiescent cells. CSCs are protected inside the niche in a dormant scheme. These different synthetic nanoparticles (NPs), such as liposomes, micelles, polymers, and gold nanoparticles, have been shown to successfully deliver anticancer drugs to the targeted CSCs by using CSC-specific markers, such as CD44, CD90, and CD133.

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
