# Peer review of "Dissecting Tumor Growth: The Role of Cancer Stem Cells in Drug Resistance and Recurrence"

_cancers, 2022, doi:10.3390/cancers14040976_

Round 1

Reviewer 1 Report

The manuscript seems to be much improved. But it still has fly in the ointment. Fig 3 and 4 are unattractive compared to other figures.

Reviewer 2 Report

The authors have addressed my concerns and the article may now be accepted for publication.

This manuscript is a resubmission of an earlier submission. The following is a list of the peer review reports and author responses from that submission.

Round 1

Reviewer 1 Report

This review paper describes the role of cancer stem cells in tumorigenesis including molecular involvement and pathological meaning.  It is well-structured and arranged. But, it seems somehow literal and monotonous. I don't feel it is differentiated from other CSC review papers. I'd strongly recommend adding drug resistance and metastasis-associated CSCs function on the current manuscript. Also the authors need to add 2 or 3 more figures and make the graphics polished.   

Reviewer 2 Report

Although I appreciated the effort of the authors, in my opinion the present review lacks of an original point of view and so the interesting for the readers could be low. 
There are several papers and reviews in this field and the paper of Aramini et al. doesn't add anything to the general discussion.
For the aforementioned, I suggest to reject the manuscript in the present form. 

Reviewer 3 Report

The manuscript provides a relatively comprehensive overview of a complex topic.

The authors discuss CSCs in relation to chemo- and radio-therapy resistance, however there is a very limited discussion of CSCs in regard to immunotherapy resistance (for example please see Front. Cell Dev. Biol., 21 June 2021 | https://doi.org/10.3389/fcell.2021.692940. Given the importance of immunotherapy in our current cancer treatments, this topic should be discussed in more detail.